# Symptoms, Symptom Profiles, and Healthcare Utilization in Patients with Hematologic Malignancies: A Retrospective Observational Cohort Study and Latent Class Analysis

**DOI:** 10.3390/curroncol32020062

**Published:** 2025-01-25

**Authors:** Reanne Booker, Richard Sawatzky, Aynharan Sinnarajah, Siwei Qi, Claire Link, Linda Watson, Kelli Stajduhar

**Affiliations:** 1Palliative Care, Arthur JE Child Comprehensive Cancer Centre, Calgary, AB T2N 4N1, Canada; 2School of Nursing, University of Victoria, Victoria, BC V8P 5C2, Canada; kis@uvic.ca; 3School of Nursing, Trinity Western University, Langley, BC V2Y 1Y1, Canada; rick.sawatzky@twu.ca; 4Centre for Advancing Health Outcomes, St. Paul’s Hospital, Vancouver, BC V6Z 1Y6, Canada; 5University of Gothenburg Centre for Person-Centred Care (GPCC), Sahlgrenska Academy, University of Gothenburg, SE-405 30 Gothenburg, Sweden; 6Division of Palliative Medicine, Queen’s University, Kingston, ON K7L 3J7, Canada; 7Surveillance & Reporting, Advanced Analytics, Cancer Research & Analytics, Cancer Care Alberta, Alberta Health Services, Calgary, AB T2N 2T9, Canada; siwei.qi@albertahealthservices.ca; 8Supportive Care Services and Patient Experience, Cancer Care Alberta, Alberta Health Services, Calgary, AB T2N 2T9, Canada; claire.link@albertahealthservices.ca (C.L.);; 9Faculty of Nursing, University of Calgary, Calgary, AB T2N 1N4, Canada

**Keywords:** hematologic malignancies, quality of life, symptom burden, symptom profiles, healthcare utilization, palliative care

## Abstract

Symptom burden is known to be high in patients with hematologic malignancies and can adversely impact patients’ quality of life. The aims of this retrospective observational cohort study were to explore symptoms in patients with hematologic malignancies, including during the last year of life, to explore symptom profiles in patients with hematologic malignancies, and to explore associations among symptoms/symptom profiles and demographic, clinical, and treatment-related variables. Symptom prevalence and severity and symptom profiles were explored in patients with hematologic malignancies who completed patient-reported outcome measures (n = 6136) between October 2019 and April 2020. Emergency department visits and hospital admissions during the study period were reviewed. Chart audits were undertaken for patients who died within a year of completing patient-reported outcome measures (n = 432) to explore symptoms and healthcare utilization in the last year of life. Patients with hematologic malignancies in this study reported multiple symptoms co-occurring, with more than 50% of patients reporting four or more symptoms. Classes of co-occurring symptoms (symptom profiles) were associated with demographic and clinical factors as well as with healthcare utilization, particularly emergency department visits. The most reported symptoms were tiredness, impaired well-being, and drowsiness. The findings emphasize the need for more supports for patients with hematologic malignancies, particularly for symptom management.

## 1. Introduction

Advances in the treatment of hematologic malignancies (HMs), such as targeted therapies, immunotherapies, and chimeric antigen receptor (CAR) T-cell therapy, have led to improvements in survival yet people with HMs still face significant risks of morbidity and mortality [1,2,3,4]. Patients with HMs may experience disease and treatment-related symptoms that can adversely impact quality of life [1,5]. Symptom burden, a concept that considers symptom prevalence, frequency, and intensity, can fluctuate throughout the disease and treatment trajectories [6]. Previous research has found that symptom burden is high in patients with HMs, with patients reporting both frequent and severe symptoms throughout the disease trajectory [7,8].

Both the underlying HM as well as the associated treatments can contribute to an array of physical and psychosocial symptoms [9]. Some HMs are considered incurable but may be responsive to treatment, whereas others are managed with observation [1,4,10]. Observation can lead to uncertainty and associated emotional distress that has been reported by some patients to be even more burdensome than physical symptoms [11]. Several studies reported that patients with HMs often experience multiple symptoms or clusters. Symptoms in a cluster may share a common underlying biological etiology or pathogenic mechanism, such as being driven by inflammatory cytokines, hormones, neurotransmitters, or other immunomodulators [12]. For example, Manitta et al. [8] found that patients with HMs (N = 180) reported a mean number of 8.8 symptoms (range 2.9–14.7). Similarly, Zimmermann et al. [13] found that patients with newly diagnosed or recently relapsed acute leukemia who had been referred to palliative care reported a median of nine physical and two psychological symptoms. Other symptoms that were prominent across many studies include: impaired well-being, pain, lack of appetite, drowsiness, tingling hands/feet, insomnia, breathlessness, anxiety, and delirium [14,15,16,17,18,19,20]. The co-occurrence of multiple symptoms has been associated with impairments in patients’ functional status and quality of life [12,21,22,23,24,25]. To date, most research on co-occurring symptoms in patients with HMs has focused on a single type of HM such as lymphoma [23,26], leukemia [27,28], or multiple myeloma [29,30,31].

Previous research has explored the association between sociodemographic factors and symptoms in patients with cancer. Studies on patients with HMs have found that being female [32,33,34], of younger age [32], lower education level [35], without a partner [35], and being unemployed [32] are associated with worse symptoms and worse quality of life. Rurality has also been found to be associated with more distress and worse physical health-related quality of life [36]. Healthcare utilization has been found to be higher among patients who reside rurally, with more emergency department (ED) visits and hospital admissions (HAs) compared to patients from urban regions [37]. Rurality has also been found to adversely impact survival after a cancer diagnosis [38,39].

There is a need to better understand symptoms, including profiles of co-occurring symptoms, in patients with HMs so that interventions can be tailored to reduce symptoms and improve quality of life. Further, considering associations among patient sociodemographic factors, clinical variables, and symptoms in patients with HMs may also contribute to the development of targeted interventions and strategies to help reduce symptoms. The aims of this study were to explore the prevalence and severity of symptoms in patients with HMs, including during the last year of life, and to explore symptom profiles in patients with HMs. Associations among symptoms/symptom profiles and demographic, clinical, and treatment-related variables were explored as were relationships among symptoms, symptom profiles, and healthcare utilization, including ED visits and HAs, for patients with HMs.

## 2. Materials and Methods

### 2.1. Study Design

In Cancer Care Alberta (CCA), patients’ symptoms are routinely measured using patient-reported outcome measures (PROMs); such data are typically collected at first consult and routine clinic and treatment visits. For this study, a secondary analysis of data previously collected for a retrospective observational cohort study was undertaken. Patients who had a diagnosis of HM and PROMs were filled out between 1 October 2019, and 1 April 2020 [40].

Data on demographic characteristics, patient-reported outcomes (PROs), and acute care utilization data were collected from the Alberta Cancer Registry, electronic medical records, the Discharge Abstract Database [41], and the National Ambulatory Care Reporting System [42]. The data were analyzed and organized by cohort where Cohort A (n = 944) consisted of patients who did not complete PROMs (“No PROMs”), Cohort B (n = 6136) consisted of patients who completed PROMs (“PROMs”), and Cohort C consisted of patients who died within a year of completing PROMs (n = 432) (“Died within a Year of PROMs”). Cohorts B (PROMs) and C (Died Within a Year of PROMs) were not mutually exclusive.

#### 2.1.1. Data Sources

The Putting Patients First (PPF) questionnaire is used at all 17 cancer care facilities in Alberta and was the PROM used to assess patients’ symptoms prior to clinic and treatment appointments. The PPF questionnaire includes the revised Edmonton Symptom Assessment System (ESAS-r) and an expanded Canadian Problem Checklist (CPC) [43,44,45]. For patients who had completed the PPF multiple times during the study period, the last ESAS-r scores collected during the study period were used. Chart audits were conducted for patients in Cohort C (Died Within a Year of PROMs).

#### 2.1.2. Edmonton Symptom Assessment System (ESAS)

The ESAS is a PROM used to assess symptoms in patients with advanced cancer [46]. A revised version of the ESAS, the ESAS-r, includes nine symptoms (pain, tiredness, breathlessness, drowsiness, well-being, nausea, anxiety, lack of appetite, and depression) with the option for patients to add an additional 10th symptom [40,47,48]. Severity of symptoms is rated on a scale of 0 to 10, with 10 indicating the highest severity [40,47].

#### 2.1.3. Sociodemographic, Disease, and Treatment-Specific Variables

We collected data about age, sex, diagnosis, Charlson comorbidity index (CCI), mean and median income, and rurality. Cancer Care Alberta uses a modified version of the CCI that excludes cancer given that all patients have a diagnosis of cancer [49]. Age was collapsed into ≤60 and >60, congruent with the World Health Organization definition of aging populations as those over 60 years [50]. At the time of ethics approval, Cancer Care Alberta had been utilizing an electronic health record system (EHR) called ARIA that captures data throughout a patient’s care trajectory, including demographic data, as well as information about the diagnosis, treatment, comorbidities, and medications used [51]. Data extracted from chart audits included marital status, reasons for ED visits and HAs, and use of supportive care medications.

#### 2.1.4. Acute Care Utilization Outcomes

Emergency department (ED) visits and HA data were extracted from the National Ambulatory Care Reporting System and the Discharge Abstract Database, respectively; ED visits and HAs within 7 days of completed PROM were included. The 7-day interval for ED visits and HAs was used as Watson et al. [40] found literature that a similar study [52] used the same interval. As per Barbera et al. [53], the 7-day window could allow for the patient’s provider to respond to the symptom scores while still potentially contributing to the ED visit. For patients who had died within 1 year of completing a PROM, reasons for ED visits and HAs were also extracted via chart audits.

### 2.2. Ethical Considerations

Ethical approval was obtained from the Health Research Ethics Board of Alberta—Cancer Committee (HREBA.CC-20-0022).

### 2.3. Data Analysis

Demographic, disease and clinical/treatment characteristics, and healthcare utilization data were summarized with means and standard deviations (continuous variables), and frequencies, medians, and ranges (categorical variables) for Cohorts A (No PROMs), B (PROMs), and C (Died Within a Year of PROMs). Chi-square analyses were undertaken to examine differences in categorical variables, including differences between patients who completed PROMs and those who did not. A value of *p* < 0.05 was used as a cutoff to indicate statistical significance. Statistical analyses were performed using SPSS (IBM, Chicago, IL, USA) Version 29 [54] and MPlus (Muthén & Muthén, Los Angeles, CA, USA) Version 8.10 [52].

#### 2.3.1. Symptom Prevalence and Severity

For symptom prevalence, the number and types of symptoms that each participant reported were assessed. For symptom severity [47,55,56], ESAS-r scores were collapsed into the following categories: none (score of 0), mild (score of 1–3), moderate (score of 4–6), and severe (score of 7–10) [47,55,56]. Descriptive statistics for ESAS-r scores were calculated. The mean number of symptoms reported by patients and the number of moderate and severe symptoms were calculated for Cohorts B (PROMs) and C (Died Within a Year of PROMs).

#### 2.3.2. Latent Class Analysis

A patient-centered approach was undertaken which categorizes subgroups of patients with distinct symptom profiles that represent the types and severity of symptoms endorsed by patients [57,58]. Latent class analysis was undertaken to look for ‘hidden’ subgroups (latent classes) of patients who have similar symptom profiles based on the prevalence and severity of each symptom. Each symptom was thus represented as an ordinal variable with values 0 (=none), 1 (=mild), 2 (=moderate), and 3 (=severe), as described above. To identify the latent classes, we started with a one-class model and then added additional classes, one at a time, until the best-fitting model was identified. We assessed model fit based on guidelines described in the literature (outlined below) and interpretability [59,60]. We used the Bayesian information criterion (BIC) to compare the relative fit of models with k and k-1 classes (lower values for the BIC indicate better fit) [23,60]. Additionally, we used the bootstrapped likelihood ratio test and the Vuong-Lo-Mendell-Rubin adjusted likelihood ratio test [23] to determine the statistical significance of the difference between k and k-1 class models. Entropy, a diagnostic statistic that is used to examine how accurately the model defines the classes, was also examined [23,60]. An entropy value of greater than 0.80 is generally recommended [60,61]. The latent classes were ordered from higher symptom burden to lower symptom burden to facilitate interpretation. However, it is important to note that symptom profiles do not directly correspond with the severity of symptom burden, but rather reflect profiles of symptom occurrence and severity.

To look for associations among categorical sociodemographic and clinical variables and latent class membership, bivariate regression analyses were undertaken using the ‘DCAT’ procedure [62,63]. DCAT is a syntax statement that is used in MPlus to analyze distal categorical outcomes following the procedures described by Lanza et al. [64] This approach factors in the possibility of participants’ partial membership (imperfect entropy) in latent classes [60,61]. Latent class analysis provides probabilities of class membership but not absolute membership in each class; it is possible for individuals to have partial membership in multiple classes [60,61]. Latent class proportions for each sociodemographic and clinical variable were estimated and the Wald Chi-square test was used to look for differences in proportions among the classes.

## 3. Results

### 3.1. Demographics

Demographic information is shown in Table 1. The full cohort (N = 7080) was comprised of 6136 patients who completed PROMs (Cohort B) and 944 patients who did not complete PROMs (Cohort A). There were more males (n = 4055, 57.3%) than females (n = 3025, 42.7%). Mean age was 64.12 years (range 18–100, standard deviation 15.16). Most patients were from urban settings (n = 5533, 78.1%) with only 21.7% (n = 1534) residing in rural regions. Most patients had CCI scores of 0 (n = 5932, 83.8%). Age, CCI, and rurality were associated with PROM completion. In addition, healthcare utilization differed among patients who completed PROMs and those who did not. These differences are displayed in Table 1.

Demographic data for Cohort C (Died Within a Year of PROMs, n = 432) are shown in Table 2. Most patients in this cohort were male (63.2%) and were older than 60 years of age (86.8%).

### 3.2. Prevalence and Severity of Symptoms

Symptom prevalence and severity for Cohorts B (PROMs) and C (Died Within a Year of PROMs) are shown in Figure 1. The most prevalent symptoms in Cohort B (PROMs) were tiredness, impaired well-being, and drowsiness; these were also the symptoms that were rated the most severe. Tiredness, impaired well-being, and drowsiness were also the most reported symptoms in Cohort C (Died Within a Year of PROMs). As with Cohort B (PROMs), tiredness was the symptom that was rated severe most often. Patients in Cohort C (Died Within a Year of PROMs) rated breathlessness, drowsiness, impaired well-being, lack of appetite, and pain as severe. In terms of number of symptoms, the mean number of symptoms for patients in Cohorts B (PROMs) and C (Died Within a Year of PROMs) was 4.3 and 5.8, respectively. Most patients in Cohort B (PROMs) reported having more than three symptoms (58.5%) and more than 20% of patients reported three or more moderate/severe symptoms (Figure 2). Most patients in Cohort C (Died Within a Year of PROMs) also reported having more than three symptoms (81.3%), and more than half (57.7%) of patients in this cohort reported having three or more moderate/severe symptoms (Figure 2).

### 3.3. Symptom Profiles

The latent class analysis identified six classes in Cohort B (PROMs) and three classes in Cohort C (Died Within a Year of PROMs). Information about the fit of the latent class models is shown in Table 3 and Table 4. For Cohort B (PROMs), even though the information criteria continued to decline when more classes were identified, the VLMR likelihood ratio test indicates that the difference in the fit of the six- and seven-class models was not statistically significant, thus suggesting a six-class model as being more defensible. The symptom profiles for the six classes are shown in Figure 3. The probability of belonging to Class 5 was highest (28.0%) while the probability of being in Class 1 was lowest (6.5%).

The symptom profile for Class 1 included prevalent and moderate/severe symptoms (all ESAS-r symptoms). The symptom profile for Class 2 included prevalent and mild/moderate symptoms. Patients in Class 3 reported prevalent, mostly mild/moderate symptoms, including anxiety and depression. Class 4 represented patients who reported prevalent, moderate/severe tiredness, impaired well-being, and drowsiness, with other symptoms being less prevalent and less severe. Patients in Class 5 reported prevalent, mostly mild physical symptoms, with less prevalent nausea and lack of appetite. The symptom profile for Class 6 included less prevalent, mild symptoms for all ESAS-r symptoms, particularly for drowsiness, nausea, lack of appetite, and depression. For all classes, tiredness was one of the most commonly reported symptoms and was often reported as moderate or severe. Nausea was the least reported symptom. Symptom profiles for each class are shown in Figure 3.

For Cohort C (Died Within a Year of PROMs), the latent class analysis found that a three-class model was the most defensible. The probability of belonging to Class 3 (less prevalent, mild/moderate physical symptoms) was highest (38.3%), followed closely by Class 1 (prevalent, moderate/severe symptoms) (37.4%), with the probability of belonging to Class 2 (prevalent, mild/moderate symptoms) the least (24.4%).

The symptom profile for Class 1 included prevalent, moderate/severe symptoms. Tiredness, impaired well-being, and drowsiness were the most reported symptoms. Nausea was the least commonly reported symptom. Patients in Class 2 reported prevalent, mild/moderate symptoms, with nausea being less prevalent. Patients in Class 3 reported less prevalent, mild/moderate physical symptoms except nausea, and less prevalent anxiety and depression. Symptom profiles for each class are shown in Figure 4.

#### Correlates of Latent Class Membership

There were several demographic and clinical differences between the subgroups (Table 5). Sex distribution varied from 33.6% females in Class 6 (less prevalent, mild symptoms) to 60.3% in Class 1 (prevalent, moderate/severe symptoms). Patients in Class 6 (less prevalent, mild symptoms) were more likely to be younger. The Charlson Comorbidity Index (CCI) score was associated with class membership, with patients in Class 1 (prevalent, moderate/severe symptoms) more likely to have higher CCI scores and patients in Class 6 (less prevalent, mild symptoms) more likely to have low CCI scores. Class membership was associated with type of HM, with Class 6 (less prevalent, mild symptoms) having the highest percentage of patients with Hodgkin Lymphoma, while Class 1 (prevalent, moderate/severe symptoms) had the highest percentage of patients with ‘other’ HMs.

In addition, healthcare utilization differed among classes with differences in any ED visits, number of ED visits, any HAs, and number of HAs. Classes 1 (prevalent, moderate/severe symptoms) and 2 (prevalent, mild/moderate symptoms) had the highest likelihood of having any ED visits (56.1% and 57.0%, respectively) and the highest percentage of 7+ ED visits (5.6%). Class 6 (less prevalent, mild symptoms) had the lowest percentage of any ED visits (28.3%) and the lowest percentage of 7+ ED visits (1.4%). Similarly, Class 1 (prevalent, moderate/severe symptoms) had the highest percentage of any HA (39.4%) and 7+ HAs (0.9%). Class 6 (less prevalent, mild symptoms) had the lowest percentage of any HA (12.5%) and 7+ HAs (0.2%).

For Cohort C (Died Within a Year of PROMs), there were no differences in demographic variables across the three latent classes (Table 6). There were differences among classes only for supportive care medication use. Class 1 (prevalent, moderate/severe symptoms) had the highest probability of supportive care medication use (62.7%) while Class 3 (less prevalent, mild/moderate physical symptoms, less prevalent anxiety/depression) had the lowest probability of supportive care medication use (38.3%).

## 4. Discussion

This cohort study examined symptom prevalence, severity, and symptom profiles associated with sociodemographic and clinical variables in patients with HMs. Notable findings include that the patients in this study reported multiple symptoms co-occurring at the time of PROM completion. In addition, patients experienced moderate/severe symptoms. The most reported symptoms were tiredness, impaired well-being, and drowsiness. Nausea was the least commonly reported symptom. These findings are similar to those reported by others [8,13].

Patients who died within a year of completing PROMs (Cohort C) reported prevalent and severe symptoms. This finding is consistent with others who have found high symptom burden during the end-of-life phase for patients with HMs. Leblanc et al. [65] found that patients reported high scores for fatigue (mean 5.2, SD 2.7) and loss of appetite (mean score 2.3, SD 2.9) and that both symptoms worsened as death approached [65]. Button et al. [66] undertook a literature review and found that the most frequently reported end-of-life signs and symptoms were pain, hematopoietic dysfunction, dyspnea, and reduced oral intake.

### 4.1. Patient-Reported Outcomes Measures

Patients who completed PROMs were less likely to have died compared to patients who did not complete PROMs. This aligns with the study by Barbera et al. [67] who found that the probability of survival was higher for patients exposed to ESAS questionnaires compared to those who were not (81.9% versus 76.4% survival at 1 year, 68.3% versus 66.1% at 3 years, and 61.9% versus 61.4% at 5 years, *p* < 0.0001). There are many potential reasons why PROM completion may be associated with improved survival. It is possible that patients who did not complete PROMs were too unwell to do so and as such, may have already been more likely to die due to advanced disease, relapse, and/or treatment-related complications [68,69]. In addition, PROM completion may lead to improved symptom management which could result in patients staying on treatment as scheduled and/or longer than patients with no PROM completion [69,70,71,72]. In our study, we found that patients who resided rurally/remotely were less likely to complete PROMs. It is possible that patients who reside rurally/remotely may have disparate access to diagnosis, treatment, and/or supportive care [38,73].

### 4.2. Rural and Remote Challenges

Patients who reside rurally and remotely in Canada face a wide array of challenges and barriers that can adversely impact disease and treatment-related outcomes, including symptom management, quality of life, and survival. Such barriers include but are not limited to the distance required to travel to/from cancer centers, transportation barriers, lack of access to specialty oncology and palliative care specialists, and sociodemographic factors [74,75]. In Canada, approximately 2% of specialists practice in rural areas [76]. Lack of access to oncology and palliative care specialists may adversely impact quality of life [77,78]. Patients with cancer who reside in rural or remote regions do not have the same access to the types of diagnostics, treatment, and supportive care services that are available in urban centers [38,79]. Reduced access to screening may mean that patients are diagnosed with later-stage disease, which may be associated with increased symptom burden compared to earlier-stage disease and may also adversely impact curability [80,81].

One potential strategy for increasing access to care for rural/remote patients with cancer is the use of virtual cancer care. Virtual care may help to mitigate issues related to access, allowing patients in rural/remote regions to access care, including specialty palliative care [82,83,84]. Remote symptom monitoring has been associated with improved symptom burden and improved quality of life [85,86,87]. As one example, Mooney et al. [87] conducted a randomized clinical trial to evaluate the impact of remote symptom monitoring on symptom burden, quality of life, and healthcare utilization. Patients were randomized to the intervention group (n = 128) or usual care (n = 124). Symptom burden was found to be lower and quality of life better for patients in the intervention group at months 1 and 2 [87]. Patients in the usual care group had more unplanned healthcare visits (43 visits) compared to patients in the intervention group (28 visits). It is important to note that both virtual cancer care and remote symptom monitoring are not without challenges, including issues pertaining to technology, availability of internet, digital health literacy, and privacy, to name a few [82,83,88,89].

### 4.3. Sex Differences and Symptoms

We found that female patients in Cohort B (PROMs) were more likely to be in the class that reported prevalent and moderate/severe symptoms (Class 1). Previous research has found that females with HMs are more likely to report symptoms compared to males. For example, in their integrative review of sex differences in quality of life and symptoms in patients with HMs that included 11 studies, Tinsley-Vance et al. [90] found that female patients were more likely to report moderate to severe symptoms of nausea, anxiety, drowsiness, poor well-being, and tiredness compared to male patients. Ebraheem et al. [91] also found that female sex was associated with higher odds of reporting many symptoms, including impaired well-being, pain, loss of appetite, anxiety, and nausea.

### 4.4. Age and Symptoms

Patients older than 60 were more likely to be in Class 4 (prevalent, moderate/severe tiredness, impaired well-being, drowsiness with less prevalent anxiety and depression) in Cohort B (PROMs). Research has been conflicting in terms of age and symptoms, with some studies reporting increased symptoms in patients who are older, thought to possibly be due to the contribution of comorbidities on symptom burden [92,93,94]. In contrast, other studies have reported that younger patients experience more symptoms, particularly more psychological symptoms such as anxiety and depression [95,96,97]. Such variability and discordance in findings could reflect the different populations being studied, including different types of cancer and different treatment contexts (cancer-directed treatment, observation/follow-up, survivorship, end-of-life care, and age cutoffs used).

### 4.5. Symptoms and Healthcare Utilization

Analysis of class membership probability also revealed that for patients in Cohort B (PROMs), Class 1 (prevalent, moderate/severe symptoms) was associated with higher percentages of both ED visits and HAs while Class 6 (less prevalent, mild symptoms) was associated with lower probabilities of ED visits and HAs. Nipp et al. [98] examined the relationship between physical and psychological symptoms and healthcare utilization in patients with advanced cancer (N = 1036) and found that high symptom burden was associated with prolonged hospital admission and readmission. Similarly, Mian et al. [18] found that among patients with multiple myeloma (N = 2876), higher total ESAS scores were associated with greater odds of ED visits/hospitalizations (Odds Ratio (OR): 1.34, 95% CI: 1.29–1.38).

### 4.6. Symptoms in the Last Year of Life

As with other studies on symptom burden in patients with HMs [99,100], we also found that patients with HMs experienced high symptom burden in the last year of life and high healthcare utilization in the last year of life, with 90% of patients having had both ED visits and HAs. The chart audits found that most ED visits had been for symptom management. Our findings align with a study by Phung et al. [101] who conducted a descriptive, retrospective study to review after-hours calls to the hematology/oncology clinic. Of 500 calls representing 398 unique patients, the authors found that most calls were to report symptoms (n = 325, 65%); 120 (24%) of the calls had been from patients with HMs. Grewal et al. [102] conducted a retrospective study to examine ED visits (n = 218,459 visits) among patients receiving chemotherapy (n = 87,555) in Ontario between 2013 and 2017. The authors reported that the top three reasons for ED visits had been fever/infection, gastrointestinal issues (nausea, vomiting, diarrhea), and pain [102].

### 4.7. Implications for Practice, Education, and Research

This study found that patients with HMs experienced prevalent and often severe symptoms and that symptoms were associated with healthcare utilization, such as ED visits, highlighting the need for more support for symptom management. Early integration of palliative care has been identified as a recommended approach to address the need for better symptom management and support in this population. Previous research has demonstrated the positive impact of integrating palliative care for patients with cancer, particularly as related to reduced symptom burden and improved quality of life [103,104,105,106]. For example, Cui et al. [103] conducted a meta-analysis of the effects of early palliative care on health-related outcomes among patients with advanced cancer. The authors included 19 studies in their review and found that early palliative care positively impacted quality of life (standard mean difference (SMD) = 0.14, 95% CI: 0.62–0.223), and improved symptom burden (SMD = 0.14, 95% CI: 0.01–0.26). Rogers et al. [104] conducted a critical evaluation of four meta-analyses of randomized clinical trials that examined palliative care in oncology. The authors found that all four meta-analyses reported improved quality of life for patients randomized to receive palliative care. Despite these known benefits, patients with HMs do not routinely receive palliative care and if they do, it is typically very late in the disease trajectory [19,105,107]. Integrating palliative care into the treatment trajectory for patients with HMs may contribute to reduced symptom burden and improved quality of life, and may also potentially reduce healthcare utilization [107,108].

Of the patients who died within a year of PROM completion, symptoms were both prevalent and severe. Specifically, pain, anxiety, and depression were reported by 66.5%, 58.5%, and 51.7% of patients, respectively. However, when medications were reviewed, only 43.3% of patients had documentation of opioid pain medication use, 6.7% had documentation of anxiolytic use, and 23.8% had documentation of antidepressant use. Other authors have reported that patients often underreport, and clinicians often underestimate symptoms [109,110,111]. In addition, some patients and clinicians expect that some degree of suffering is unavoidable [111,112].

Additional reasons for patients not reporting their symptoms include patients not wanting their treatment to be interrupted or discontinued or feeling as if their symptoms were not severe enough to warrant mention to their healthcare providers [113]. Regarding pain management, patients and clinicians may harbor negative perceptions of opioid use [114,115].

Similar to physical symptoms, clinicians may underrecognize and/or undertreat psychological symptoms [116]. Walker et al. [117] analyzed data from patients with cancer who participated in routine screening for depression in Scotland (N = 21,151). The authors found that of the patients who had been diagnosed with major depression and had treatment information available (n = 1538), most were not receiving any treatment (n = 1130, 73%) and only 24% (n = 370) were taking an antidepressant [117]. Fisch et al. [118] reported similar findings in their study involving ambulatory patients with solid tumor cancers (N = 3106), where 47% (n = 1457) of patients were found to have depressive symptoms. Antidepressants had been prescribed in 25% of patients with depressive symptoms [118].

The low rate of psychotropic medications, specifically anxiolytics and antidepressants, in our study may have been due to a lack of mental health assessment, reluctance on behalf of oncology clinicians to prescribe psychotropics, concern about side effects, concern about drug-drug interactions, patient preferences (to not embark on pharmacotherapy), or possibly, because patients had access to other modalities of care, such as counseling/psychotherapy that were effective [119,120,121]. The American Society of Clinical Oncology guideline for the management of anxiety and depression in cancer survivors calls for a stepped approach that includes education, psychotherapy, and pharmacotherapy tailored to the individual patient’s needs [116].

Patient education on cancer and treatment-related symptoms as well as systematic assessment of symptoms may help to improve symptom detection and management [122,123,124]. The routine use of PROMs may also help clinicians recognize and better manage symptoms that are bothersome to patients [125,126].

A better understanding of disease and treatment-related side effects and symptoms and their underlying pathogenic mechanisms are needed [58,126]. Symptoms may co-occur in patients with HMs and further research may reveal shared pathologic processes that contribute to both the underlying disease as well as to the development of symptoms [127,128]. As one example, interleukin-6 (IL-6) is implicated in the pathogenesis of multiple myeloma [129] and also, in the development of symptoms such as fatigue [130,131,132,133]. Patton et al. [134] undertook a systematic review that included 20 studies (n = 1806 patients, n = 199 controls) to examine the literature on the relationships between cytokines and symptoms in people with incurable cancer. The authors found that symptoms experienced by patients with incurable cancer—most often, depression, fatigue, pain, and lack of appetite—were associated with circulating cytokines such as IL-6, interleukin 8, and tumor necrosis factor-alpha, among others [134].

Further research is also needed on pharmacologic and non-pharmacologic interventions for the most prevalent and severe symptoms experienced by patients with HMs. A better understanding of the pathophysiology of chemotherapy-induced nausea and vomiting (CINV) has led to the development of antiemetics such as 5-hydroxytryptamine 3 (5-HT3) receptor antagonists and neurokinin-1 receptor antagonists, which have reduced the incidence of CINV [135,136]. Unfortunately, fewer options are available to help manage symptoms such as fatigue and drowsiness. More research is needed on interventions that might help alleviate symptoms such as fatigue, impaired well-being, and drowsiness [123,125].

### 4.8. Limitations

There are some important limitations in our study. Given that not all patients completed PROMs at multiple time points, we elected to use only one time point for PROM scores. Tracking patient symptom scores over time would have allowed for a more complete picture of the symptom experience. Patients with HMs may experience rapid deterioration and as such, we may have missed important symptom scores or, conversely, may have captured symptom scores at a time when patients were particularly unwell. There were several potentially relevant variables that were not available for our study, including educational level, employment status, preferred language, and stage of disease. An additional limitation is that the ESAS-r does not capture all the symptoms and concerns that are important for patients with HMs, such as peripheral neuropathy or bowel concerns.

An additional potential confounder is that some of the data collection occurred during the early stages of the COVID-19 pandemic, particularly in March–April 2020. The first case of COVID-19 in Alberta was announced in early March 2020 with restrictions on gatherings coming into effect on 12 March 2020 [137]. Cancer Care Alberta rapidly implemented virtual clinic visits for many patients in March 2020 [138]. This shift to virtual care, in addition to the additional stressors of the pandemic, may have influenced patients’ symptom experiences, PROM completion, and results.

## 5. Conclusions

This retrospective observational study contributes to the literature on symptom burden in patients with HMs and confirms that patients with HMs experience significant symptom burden, particularly in the last year of life. In addition, this study found that symptoms and symptom profiles are associated with demographic factors as well as with healthcare utilization, particularly ED visits. This study also yields important information regarding patient groups who may benefit from more targeted symptom management interventions, such as female patients and older adults. There is a growing body of evidence demonstrating that patients with HMs may benefit from the integration of palliative care, particularly as related to reduced symptom burden and improved quality of life. Future research should continue to examine how best to integrate palliative care into the care of patients with HMs.

## Figures and Tables

**Figure 1 curroncol-32-00062-f001:**
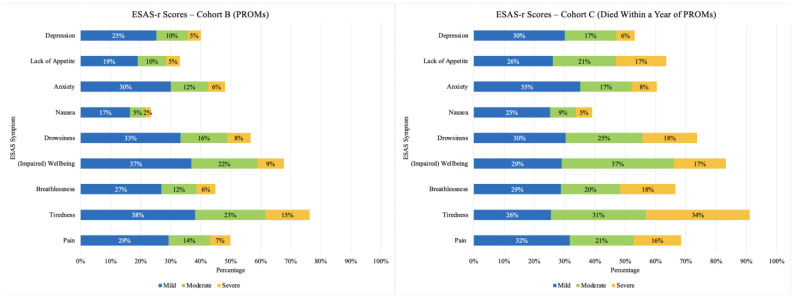
Symptom Prevalence and Severity—Cohorts B (PROMs) and C (Died Within a Year of PROMs). Note. For each bar, the stacks represent symptom severity as ‘mild’, ‘moderate’, and ‘severe’, and the total length of each bar represents the overall prevalence (i.e., whether the symptom occurred or not). To improve visualization of symptom prevalence, the category ‘none’ is not shown.

**Figure 2 curroncol-32-00062-f002:**
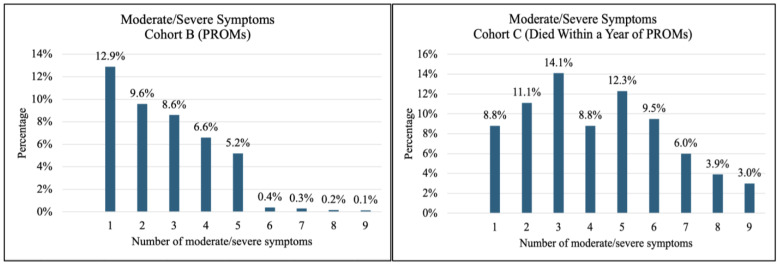
Number of Moderate and Severe Symptoms—Cohorts B (PROMs) and C (Died Within a Year of PROMs).

**Figure 3 curroncol-32-00062-f003:**
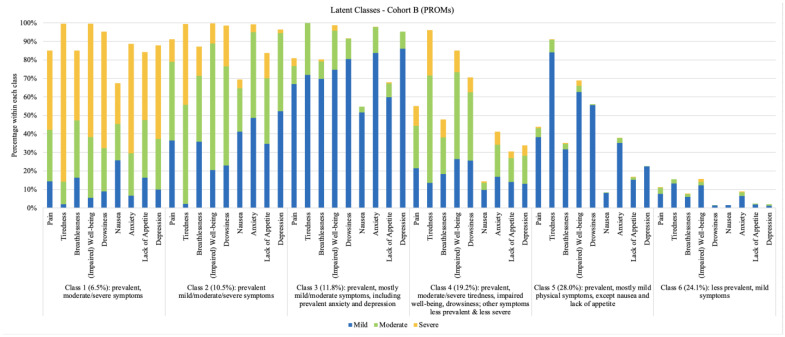
Latent Classes, Cohort B (PROMs). Note. For each bar, the stacks represent symptom severity as ‘mild’, ‘moderate’, and ‘severe’, and the total height for each bar represents the overall prevalence (i.e., whether the symptom occurred or not). To improve visualization of symptom prevalence, the category ‘none’ is not shown.

**Figure 4 curroncol-32-00062-f004:**
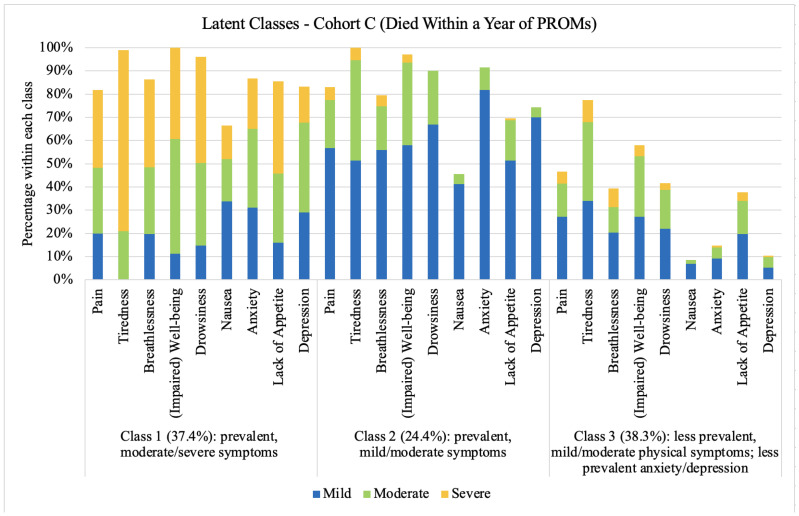
Latent Classes, Cohort C (Died Within a Year of PROMs). Note. For each bar, the stacks represent symptom severity as ‘mild’, ‘moderate’, and ‘severe’, and the total height for each bar represents the overall prevalence (i.e., whether the symptom occurred or not). To improve visualization of symptom prevalence, the category ‘none’ is not shown.

**Table 1 curroncol-32-00062-t001:** Demographic Data, Cohorts A (No PROMs) and B (PROMs).

Variable		Cohort A: No PROMS	Cohort B: PROMs	*p*
		N = 944 (%)	N = 6136 (%)	
Age	≤60	352 (37.3)	2084 (34.0)	0.045
	>60	592 (62.7)	4052 (66.0)	
Sex	Male	513 (54.3)	3542 (57.7)	0.051
	Female	431 (45.7)	2594 (42.3)	
Type of HM	Leukemia	261 (27.6)	1605 (26.2)	<0.001
	HL	56 (5.9)	373 (6.1)	
	NHL	357 (37.8)	2260 (36.8)	
	MM	84 (8.9)	1022 (16.7)	
	IP diseases	48 (5.1)	340 (5.5)	
	Other	138 (14.6)	535 (8.7)	
CCI ^a^	0	732 (77.5)	5200 (84.7)	<0.001
	1	99 (10.5)	524 (8.5)	
	≥2	113 (12.0)	412 (6.7)	
Rurality	Rural	237 (25.2)	1297 (21.2)	0.006
	Urban	705 (74.8)	4828 (78.8)	
ED Visits	None	499 (52.9)	3632 (59.2)	<0.001
Number of ED Visits ^a^	1–3	342 (36.2)	2044 (33.3)	
	4–6	59 (6.3)	302 (4.9)	
	7+	44 (4.7)	158 (2.6)	
Hospital Admissions	None	680 (72.0)	4691 (76.5)	<0.001
Number of HAs ^a^	1–3	228 (24.2)	1283 (20.9)	
	4–6	27 (2.9)	144 (2.3)	
	7+	9 (1.0)	18 (0.3)	
Deceased	No	829 (87.8)	5691 (92.7)	<0.001
	Yes	115 (12.2)	445 (7.3)	

Note. Due to missing data, the frequencies do not add up to the total sample size for some of the variables. The percentages are out of the number of participants who did not have missing data for the corresponding variable. *p* = *p*-value based on a chi-square test. HM = hematologic malignancy, HL = Hodgkin lymphoma, NHL = non-Hodgkin lymphoma, MM = multiple myeloma, CCI = Charlson Comorbidity Index, ED = emergency department, HA = hospital admission. ^a^ Count variables were discretized to facilitate interpretation.

**Table 2 curroncol-32-00062-t002:** Demographic Data, Cohort C (Died Within a Year of PROMs).

Variable		N = 432	%
Age	≤60	57	13.2
	>60	375	86.8
Sex	Male	273	63.2
	Female	159	36.8
Type of HM	Acute leukemia	61	14.1
	Chronic leukemia	39	9.0
	Lymphoma	177	41.0
	Multiple myeloma	107	24.8
	Myelodysplastic syndrome	34	7.9
	Other	14	3.2
CCI	0	269	62.3
	1	90	20.8
	≥2	73	16.9
Rurality	Urban	332	76.9
	Rural	100	23.1
Marital Status	Not married	148	34.3
	Married	284	65.7
ED Visits	None	45	10.4
Number of ED Visits ^a^	1–3	256	59.3
	4–6	87	20.1
	7+	44	10.2
Reasons for ED Visits	Symptom management	207	47.9
	Disease/treatment	33	7.6
	Complications	111	25.7
	Other	35	8.1
	n/a	46	10.6
Hospital Admissions	None	43	10.0
Number of HAs ^a^	1–3	326	75.5
	4–6	56	13.0
	7+	7	1.6
Reasons for HAs	Symptom management	50	11.6
	Disease/treatment	9	2.1
	Complications	292	67.6
	Other	38	8.8
	n/a	43	10.0
Bone Marrow/Stem Cell Transplant	No	358	82.9
	Yes	74	17.1
Lines of Treatment ^a^	None	54	12.5
	1–3	247	57.2
	4–6	95	22.0
	7+	36	8.3
Supportive Care Medications	No	171	39.6
	Yes	261	60.4
	Opioid pain medication	187 ^b^	43.3
	Non-opioid pain medication	97	22.5
	Anxiolytic	29	6.7
	Antidepressant	103	23.8
Progressive Disease ^c^	No	253	58.6
	Yes	179	41.4
Place of Death	Hospital	278	64.4
	Home	82	19.0
	Hospice	51	11.8
	Other	12	2.8
	Unknown	9	2.1

Note. Due to missing data, the frequencies do not add up to the total sample size for some of the variables. The percentages are out of the number of participants who did not have missing data for the corresponding variable. HM = hematologic malignancy, CCI = Charlson Comorbidity Index, ED = emergency department, HA = hospital admission, n/a = not available. ^a^ Count variables were discretized to facilitate interpretation. ^b^ Some patients had multiple types of supportive care medications; ^c^ Documented progressive, relapsed, recurrent, or refractory.

**Table 3 curroncol-32-00062-t003:** Latent Classes, Cohort B (PROMs).

k	BIC	Entropy	VLMR	BLRT	P: Class 1	P: Class 2	P: Class 3	P: Class 4	P: Class 5	P: Class 6	P: Class 7
1	118,983.50	n/a	n/a	n/a	1						
2	104,160.70	0.86	*p* < 0.01	*p* < 0.01	0.44	0.56					
3	99,493.40	0.83	*p* < 0.01	*p* < 0.01	0.26	0.40	0.34				
4	97,730.51	0.84	*p* < 0.01	*p* < 0.01	0.14	0.25	0.38	0.23			
5	96,279.90	0.83	*p* < 0.01	*p* < 0.01	0.14	0.25	0.29	0.13	0.20		
6	95,792.97	0.83	*p* < 0.01	*p* < 0.01	0.12	0.24	0.19	0.07	0.11	0.28	
7	95,402.23	0.82	*p* = 0.70	*p* < 0.01	0.08	0.11	0.25	0.08	0.12	0.21	0.15

Note. BIC: Bayesian Information Criterion; VLMR: Vuong-Lo-Mendell-Rubin Likelihood Ratio Test *p*-value comparing k and k-1 class models; BLRT: Bootstrapped likelihood ratio test *p*-value comparing k and k-1 class models; P: Probability of latent class membership predicted by the model (for some models, the probabilities do not add exactly to 1 due to rounding).

**Table 4 curroncol-32-00062-t004:** Latent Classes, Cohort C.

k	BIC	Entropy	VLMR	BLRT	P: Class 1	P: Class 2	P: Class 3	P: Class 4
1	9798.99	n/a	n/a	n/a	1			
2	9014.76	0.87	*p* < 0.01	*p* < 0.01	0.44	0.56		
3	8842.29	0.86	*p* = 0.01	*p* < 0.01	0.24	0.37	0.38	
4	8835.93	0.86	*p* = 0.78	*p* < 0.01	0.32	0.24	0.21	0.23

Note. BIC: Bayesian Information Criterion; VLMR: Vuong-Lo-Mendell-Rubin Likelihood Ratio Test *p*-value comparing k and k-1 class models; BLRT *p*-value: Bootstrapped likelihood ratio test *p*-value comparing k and k-1 class models; P: Probability of latent class membership predicted by the model n/a: not applicable. For some models, the probabilities do not add exactly to 1 due to rounding.

**Table 5 curroncol-32-00062-t005:** Correlates of Class Membership, Cohort B (PROMs).

Variable	Class 1 Prevalent, Moderate/ Severe Symptoms	Class 2 Prevalent, Mild/ Moderate Symptoms	Class 3 Prevalent, Mostly Mild/Moderate Symptoms, Including Anxiety/ Depression	Class 4 Prevalent, Moderate/ Severe Tiredness, Impaired Well-Being, Drowsiness	Class 5 Prevalent, Mostly Mild Physical Symptoms Except Nausea, Lack of Appetite	Class 6 Less Prevalent, Mild Symptoms	x^2^
	n = 397 (6.5%) %	n = 639 (10.5%) %	n = 720 (11.8%) %	n = 1174 (19.2%) %	n = 1707 (28.0%) %	n = 1470 (24.1%) %	*p*
Sex							
Male	39.7	54.8	62.5	51.5	58.0	66.4	101.66
Female	60.3	45.2	37.5	48.5	42.0	33.6	*p* < 0.001
Age							
≤60	30.3	32.2	35.4	28.9	34.6	38.6	25.11
>60	69.7	67.8	64.6	71.1	65.4	61.4	*p* < 0.001
Type of HM							
Leukemia	4.4	3.9	4.1	5.2	6.4	8.7	90.32
HL	35.5	35.3	35.1	36.4	37.4	38.4	*p* < 0.001
NHL	24.6	24.4	22.9	27.6	26.0	27.9	
MM	16.5	19.9	24.0	14.8	18.0	11.6	
IP	6.8	6.6	5.6	6.4	4.4	5.5	
Other	12.2	10.0	8.3	9.6	7.7	7.8	
CCI							
0–1	86.4	87.2	94.0	92.2	95.5	95.9	64.72
2+	13.6	12.8	6.0	7.8	4.5	4.1	*p* < 0.001
Rurality							
Urban	78.3	80.0	81.4	78.0	78.2	78.5	3.56
Rural	21.7	20.0	18.6	22.0	21.8	21.5	*p* = 0.62
ED Visits							
No	43.9	43.0	55.8	51.5	65.3	71.7	235.28
Yes	56.1	57.0	44.2	48.5	34.7	28.3	*p* < 0.001
Number of ED Visits							
None	44.0	43.0	55.8	51.4	65.3	71.7	252.70
1–3	41.2	43.6	37.1	38.8	29.5	24.5	*p* < 0.001
4–6	9.2	8.4	5.2	6.4	3.6	2.4	
7+	5.6	5.0	1.9	3.3	1.6	1.4	
Hospital Admissions							
No	60.6	61.9	72.5	73.6	80.1	87.5	223.42
Yes	39.4	38.1	27.5	26.4	19.9	12.5	*p* < 0.001
Number of HAs							
None	60.5	62.0	72.5	73.7	80.0	87.5	230.43
1–3	31.9	33.1	25.1	23.1	18.3	11.5	*p* < 0.001
4–6	6.8	4.7	2.2	2.8	1.5	0.8	
7+	0.9	0.3	0.3	0.4	0.2	0.2	
Deceased							
No	94.1	91.6	93.2	91.6	92.6	93.7	5.01
Yes	5.9	8.4	6.8	8.4	7.4	6.3	*p* = 0.415

Notes. HM = hematologic malignancy, HL = Hodgkin lymphoma, NHL = non-Hodgkin lymphoma, MM = multiple myeloma, IP = immunoproliferative disorder, CCI = Charlson Comorbidity Index, ED = emergency department, HA = hospital admission. Blue and red cells indicate the classes with the lowest and highest prevalence, respectively.

**Table 6 curroncol-32-00062-t006:** Correlates of Class Membership, Cohort C (Died Within a Year of PROMs).

Variable	Class 1 Prevalent, Moderate/ Severe Symptoms	Class 2 Prevalent, Mild/ Moderate Symptoms	Class 3 Less Prevalent Mild/Moderate Physical Symptoms; Less Prevalent Anxiety, Depression	x^2^
	n = 161 (37.4%) %	n = 105 (24.4%) %	N = 165 (38.3%) %	*p*
Sex				
Male	58.3	69.0	63.7	2.10
Female	41.7	31.0	36.3	*p* = 0.350
Age				
≤60	15.9	8.8	13.5	1.35
>60	84.1	91.2	86.5	*p* = 0.510
Type of HM				
Acute Leukemia	10.9	16.1	16.4	11.95
Chronic Leukemia	11.6	6.8	7.8	*p* = 0.289
Lymphoma	37.5	36.5	47.2	
Multiple Myeloma	25.3	32.8	19.6	
Myelodysplastic Syndrome	11.0	7.0	4.7	
Other	3.0	0.7	4.3	
Rurality				
Urban	75.6	76.1	78.2	0.28
Rural	24.4	23.9	21.8	*p* = 0.870
Marital Status				
Not Married	36.6	26.1	36.8	2.03
Married	63.4	73.9	63.2	*p* = 0.363
ED Visits				
No	10.5	12.0	8.3	0.78
Yes	89.5	88.0	91.7	*p* = 0.677
Number of ED Visits				
None	10.4	13.0	8.3	5.99
1–3	61.0	55.9	60.4	*p* = 0.424
4–6	16.0	19.4	24.3	
7+	12.6	11.7	7.0	
HAs				
No	10.6	7.8	10.8	0.47
Yes	89.4	92.2	89.2	*p* = 0.791
Number of HAs				
None	10.6	7.8	10.8	1.32
1–3	72.8	80.4	74.6	*p* = 0.970
4–6	15.0	9.8	13.1	
7+	1.6	1.9	1.4	
Supportive Care Medications				
No	37.3	55.6	61.7	17.74
Yes	62.7	44.4	38.3	*p* < 0.001
Progressive Disease				
No	60.8	62.5	54.5	1.40
Yes	39.2	37.5	45.5	*p* = 0.496

Notes. HM = hematologic malignancy, ED = emergency department, HA = hospital admission. Blue and red cells indicate the classes with the lowest and highest prevalence, respectively.

## Data Availability

The data used in this study are not publicly available. However, data may be provided upon request, on a case-by-case basis. Please contact the corresponding author.

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
