# Peer review of "Symptoms, Symptom Profiles, and Healthcare Utilization in Patients with Hematologic Malignancies: A Retrospective Observational Cohort Study and Latent Class Analysis"

_curroncol, 2025, doi:10.3390/curroncol32020062_

Round 1

Reviewer 1 Report

Comments and Suggestions for Authors

The authors should be congratulated for conducting this research with such a large sample of patients. However, numerous weaknesses dampen this reviewer's enthusiasm for this paper.

First and foremost, the paper lacks conceptual clarity. This paper did NOT evaluate symptom clusters. Symptom clusters are evaluated using a statistical procedure like exploratory factor analysis. Using latent variable modeling, the authors created groups of patients with distinct symptom profiles. The authors need to read Harris - PMID35502915 - to be able to revise this paper using the correct language and avoid conceptual confusion.

This review suggests that the authors read several papers that report the results of  latent variable modeling (e.g., numerous papers by Miaskowski and colleagues) to be able to determine how to present their results in a clinically meaningful way.

The classes should be ordered and named from lowest to highest symptom burden. This approach facilitates the interpretation of the study findings. It appears from the presentation of the results, that the authors did not recode the classes. In addition, the classes should have clinically meaningful names - not class numbers.

The cohorts should be named with clinically meaningful names not letters.

It is not clear what variable was used to create the latent classes. Were the classes created using symptom occurrence? This information needs to be provided in the analysis section of the paper.

Please use patient throughout the paper.

Please remove all of the percent signs on Tables 1 and 2. Simply label the column.

On table 1, the mean number of symptoms should be reported for each group of patients

For number of ED visits and HAs, were Kruskal Wallis tests run?

On figure 2, the y-axis needs a label

For the fit indices on tables 3 and 4, the values for BLRT should be reported as well as p-values. It is not clear what the other numbers are on these tables.

Figures 3 and 4 are very challenging to interpret. If the latent classes were determined using occurrence data - then the figures should be plotted differently. See papers by Miaskowski and colleagues for how to plot the figures - so that they can be interpreted more easily.

Add number of symptoms per class to tables 5 and 6.

On tables 5 and 6, post hoc contrasts need to be reported.

Place column headers on tables 5 and 6 and remove all of the % signs.

On tables 5 and 6, ED visits and HAs need Kruskal Wallis tests.

The Discussion section contains a large amount of information that is redundant with the results. In addition, Cohorts B and C were not compared directly, so the sentences in the discussion that compare the two groups warrant revision.

As written, the Discussion section lacks clinical meaning. It requires major revisions to be clinically useful.

Author Response

We are most grateful to the reviewers for their time and for their thoughtful comments and suggestions. We have endeavoured to incorporate the feedback of both reviewers into our revised manuscript. We have responded to each of the reviewers’ comments and suggestions below.

Comments and Suggestions for Authors

The authors should be congratulated for conducting this research with such a large sample of patients. However, numerous weaknesses dampen this reviewer's enthusiasm for this paper.

First and foremost, the paper lacks conceptual clarity. This paper did NOT evaluate symptom clusters. Symptom clusters are evaluated using a statistical procedure like exploratory factor analysis. Using latent variable modeling, the authors created groups of patients with distinct symptom profiles. The authors need to read Harris - PMID35502915 - to be able to revise this paper using the correct language and avoid conceptual confusion. This review suggests that the authors read several papers that report the results of latent variable modeling (e.g., numerous papers by Miaskowski and colleagues) to be able to determine how to present their results in a clinically meaningful way.

Response: Thank you for pointing this out. We used the term symptom clusters to relate our work to that of others who have similarly operationalized symptom clusters in terms of latent classes (i.e., latent subpopulations). Given the more recently emerging conceptual and operational definitions of symptom clusters, we agree that the term symptom profiles is a better fit. We have changed this throughout the manuscript. 

The classes should be ordered and named from lowest to highest symptom burden. This approach facilitates the interpretation of the study findings. It appears from the presentation of the results, that the authors did not recode the classes. In addition, the classes should have clinically meaningful names - not class numbers.

Response: We have reordered the classes from highest to lowest symptom burden (this resulted in fewer changes than reordering from lowest to highest symptom burden and we feel that it still helps with clinical interpretability). However, if Reviewer 1 feels strongly that it should be lowest to highest, we can make this change.
Classes are named with clinically meaningful names in the figures and we have added these names in text.

The cohorts should be named with clinically meaningful names not letters.

Response: We have added names to describe the cohorts.  

It is not clear what variable was used to create the latent classes. Were the classes created using symptom occurrence? This information needs to be provided in the analysis section of the paper.

Response: The latent classes were modeled based on the ordinal distributions of the discretized ESAS variables. We have now added the following information to clarify this: “Latent class analysis was undertaken to look for ‘hidden’ subgroups (latent classes) of patients who have similar symptom profiles based on the prevalence and severity of each symptom. Each symptom was thus represented as an ordinal variable with values 0 (= none), 1 (=mild), 2 (=moderate), and 3 (= severe), as described above.”

Please use patient throughout the paper.

Response: Thank you for pointing this out. We agree and have made this change.

Please remove all of the percent signs on Tables 1 and 2. Simply label the column.

Response: Thank you for this suggestion. We have made this revision.

On table 1, the mean number of symptoms should be reported for each group of patients.

Response: Table 1 compares the demographic characteristics of the two cohorts. Instead of providing the mean numbers of symptoms, we describe the prevalence and severity as well as the number of moderate or severe symptoms of symptoms for each cohort in Figures 1 and 2, respectively. A statistical comparison of symptom prevalence or severity across the two cohorts was not conducted, given that this was not the aim of our study. In addition, patients in Cohort A did not have any symptom scores as they did not complete any PROMs. 

For number of ED visits and HAs, were Kruskal Wallis tests run?

Response: We did not run Kruskal Wallis tests. Because of the distributions of these variables, and to facilitate interpretation, we discretized these variables into 3 categories (1-3, 4-6, and 7+) and conducted a chi-square test instead. We added the following comment to Tables 1 and 2 to clarify this: “Count variables were discretized to facilitate interpretation.”

On figure 2, the y-axis needs a label

Response: Thank you. This has now been added.

For the fit indices on tables 3 and 4, the values for BLRT should be reported as well as p-values. It is not clear what the other numbers are on these tables.

Response: We added a note to the table to clarify that both the BLRT and VMLR LRT statistical methods for comparing k and k-1 class models. Given that each method imposes different assumptions, we followed and cited recommendations to report both results when reporting latent class analysis results. We also followed common practice to only report the p-values for these tests, given that the likelihood ratios do not follow a conventional chi-square distribution and are therefore not readily interpretable.

Figures 3 and 4 are very challenging to interpret. If the latent classes were determined using occurrence data - then the figures should be plotted differently. See papers by Miaskowski and colleagues for how to plot the figures - so that they can be interpreted more easily.

Response: We are familiar with the papers by Miaskowski and cited several of them. However, our analyses are different in that we modeled both symptom prevalence and severity at the same time (with each symptom represented as an ordinal variable). We therefore could not report the figures in the same way. Instead, we report the figures as stacked bar graphs (following an approach similar to other latent class analyses of ordinal variables). We added the following note to each figure to clarify this: “For each bar, the stacks represent symptom severity as ‘mild’, ‘moderate’, and ‘high’, and the total height for each bar represents the overall prevalence (i.e., whether the symptom occurred or not). To improve visualization of symptom prevalence, the category ‘none’ is not shown.”

Add number of symptoms per class to tables 5 and 6.

Response: Given that we characterized each class based on the prevalence and severity of each symptom, we do not find that the number of symptoms would add further meaningful information.  

On tables 5 and 6, post hoc contrasts need to be reported.

Response: Thank you for this suggestion. We feel that adding in the post hoc comparisons would make the reporting complicated and potentially difficult to interpret. We have instead followed the example of Wallstrom et al. (2022) who highlighted the highest and lowest percentage for each variable within each class.  

Place column headers on tables 5 and 6 and remove all of the % signs.

Response: Thank you, we have made these changes.

On tables 5 and 6, ED visits and HAs need Kruskal Wallis tests.

Response: See our response to the above comment. Because of the distributions of these variables, and to facilitate interpretation, we discretized these variables into 3 categories (1-3, 4-6, and 7+) and conducted a chi-square test instead.

The Discussion section contains a large amount of information that is redundant with the results. In addition, Cohorts B and C were not compared directly, so the sentences in the discussion that compare the two groups warrant revision.

Response: Thank you. We have tried to make the Discussion section less redundant with the results section. We have removed any comparisons between Cohorts B and C.

As written, the Discussion section lacks clinical meaning. It requires major revisions to be clinically useful.

Response: We have added more content to the Discussion section that we hope adds clinical meaning.

Reviewer 2 Report

Comments and Suggestions for Authors

Manuscript Title: Symptoms, symptom clusters, and healthcare utilization in patients with hematologic malignancies: a retrospective cohort study and latent class analysis

Authors: Reanne Booker, Richard Sawatzky, Aynharan Sinnarajah, Siwei Qi, Claire Link, Linda Watson, Kelli Stajduhar

General Comments

This paper addresses an important topic, the high symptom burden among patients with hematologic malignancies, particularly during their last year of life. It effectively highlights the critical need for enhanced symptom management strategies, presenting a compelling case for targeted interventions, especially for specific demographic groups such as older adults and females. The study also provides valuable insights into the relationship between symptoms, symptom clusters, demographic factors, and healthcare utilization, reinforcing its significance in advancing supportive care for this population.

The research design is appropriate for the study objectives, and the use of validated tools like the ESAS-r and CPC ensures the reliability of the findings. The incorporation of robust data sources, including the Alberta Cancer Registry, electronic medical records, and national healthcare databases, further strengthens the study's credibility. The statistical analyses, including latent class analysis, are rigorous and well-suited to the research questions, offering meaningful insights into distinct symptom profiles.

Strengths

Relevance and Significance:

* The paper highlights a critical issue in hematologic malignancies, underscoring the high symptom burden and the need for better symptom management approaches.

Methodology:

* The study's design, a secondary data analysis of a retrospective cohort, is well-justified.

* The use of validated tools like ESAS-r and CPC enhances data reliability.

* Comprehensive data sources ensure robust findings.

Analysis:

* The statistical methods, particularly latent class analysis, are rigorous and appropriate, providing nuanced insights into symptom clusters.

* Limitations, such as the impact of the COVID-19 pandemic and the use of a single time point for PROM scores, are transparently acknowledged.

Suggestions for Improvement

Clinical Implications:

While the discussion highlights the need for improved symptom management, it could be expanded with practical examples:

* Interventions: Discuss the potential role of integrating palliative care, patient education programs, or telehealth-based symptom monitoring.

* Supportive Care Medications: Address underutilization of medications for pain, anxiety, and depression. Exploring barriers to their use and strategies to improve prescribing practices could enhance the paper's practical relevance.

Biological Mechanisms:

* The paper mentions shared pathologic processes underlying symptom clusters but could benefit from a deeper exploration of these mechanisms. Expanding on this aspect may contribute to the development of targeted interventions.

Rural Disparities:

* The study identifies that rural patients are less likely to complete PROMs but does not delve deeply into the implications. A detailed discussion on potential barriers to healthcare access in rural areas, such as transportation challenges, lack of specialized services, and geographical distance, would add depth to the findings.

Future Research Directions:

* The authors might consider a dedicated section outlining areas for further study.

Author Response

We are most grateful to the reviewers for their time and for their thoughtful comments and suggestions. We have endeavoured to incorporate the feedback of both reviewers into our revised manuscript. We have responded to each of the reviewers’ comments and suggestions below.

General Comments

This paper addresses an important topic, the high symptom burden among patients with hematologic malignancies, particularly during their last year of life. It effectively highlights the critical need for enhanced symptom management strategies, presenting a compelling case for targeted interventions, especially for specific demographic groups such as older adults and females. The study also provides valuable insights into the relationship between symptoms, symptom clusters, demographic factors, and healthcare utilization, reinforcing its significance in advancing supportive care for this population.

The research design is appropriate for the study objectives, and the use of validated tools like the ESAS-r and CPC ensures the reliability of the findings. The incorporation of robust data sources, including the Alberta Cancer Registry, electronic medical records, and national healthcare databases, further strengthens the study's credibility. The statistical analyses, including latent class analysis, are rigorous and well-suited to the research questions, offering meaningful insights into distinct symptom profiles.

Response: Thank you for this feedback.

Strengths

Relevance and Significance:

* The paper highlights a critical issue in hematologic malignancies, underscoring the high symptom burden and the need for better symptom management approaches.

Methodology:

* The study's design, a secondary data analysis of a retrospective cohort, is well-justified.

* The use of validated tools like ESAS-r and CPC enhances data reliability.

* Comprehensive data sources ensure robust findings.

Analysis:

* The statistical methods, particularly latent class analysis, are rigorous and appropriate, providing nuanced insights into symptom clusters.

* Limitations, such as the impact of the COVID-19 pandemic and the use of a single time point for PROM scores, are transparently acknowledged.

Suggestions for Improvement

Clinical Implications:

While the discussion highlights the need for improved symptom management, it could be expanded with practical examples:

* Interventions: Discuss the potential role of integrating palliative care, patient education programs, or telehealth-based symptom monitoring.

* Supportive Care Medications: Address underutilization of medications for pain, anxiety, and depression. Exploring barriers to their use and strategies to improve prescribing practices could enhance the paper's practical relevance.

Biological Mechanisms:

* The paper mentions shared pathologic processes underlying symptom clusters but could benefit from a deeper exploration of these mechanisms. Expanding on this aspect may contribute to the development of targeted interventions.

Rural Disparities:

* The study identifies that rural patients are less likely to complete PROMs but does not delve deeply into the implications. A detailed discussion on potential barriers to healthcare access in rural areas, such as transportation challenges, lack of specialized services, and geographical distance, would add depth to the findings.

Response: Thank you very much for these excellent suggestions. We have added more content to address these topics in more depth.

Future Research Directions:

* The authors might consider a dedicated section outlining areas for further study.

Response: Thank you for this suggestion. We have added some content on areas for further study.

Round 2

Reviewer 1 Report

Comments and Suggestions for Authors

The authors have been partially responsive in their revision of this manuscript. As stated previously, this paper is an important one - but several areas exist that warrant careful consideration to make this paper easy to read and clinically meaningful.

1. The paragraph that was added to the introduction (lines 47 to 59) contains incorrect information. This paragraph really is not necessary. This paper is NOT evaluating symptom clusters [i.e., how symptoms group together]. It is evaluating how PATIENTS group together. The last two sentences of this paragraph are confusing and not stated correctly.

2.  Lines 184 to 187 are not clear. In order for these sentences to have clarity, the authors need to define what they mean by symptom burden. The assumption of latent variable modeling - is that you are finding patients with distinct symptom profiles - what do the authors mean by symptom burden in this context? How did they determine symptom burden to be able to substantiate this sentence in the methods section of the paper?

3. This reviewer suggested that the cohorts and the profiles have clinical names. However, none of this information was changed in the tables or figures. Use the clinical names throughout the paper and on all of the tables and figures.

4. On the figure legends the groups are called mild, moderate, and high. However, on the figures and in the text, the groups are not high but severe. Please correct these errors.

5. The legends state that the prevalence is plotted (i.e., whether the symptom occurred or not). You cannot plot what has not occurred.

6. In the results section, the authors name the classes using the terms frequent and severe. However, they did not assess frequency. This study assessed symptom occurrence and severity. The other dimensions of the symptom experience that were not assessed were distress and frequency (i.e., how often the symptom occurs within a specified time interval (e.g., daily, weekly, monthly). This language needs to be corrected.

7. The shading on Tables 5 and 6 are not interpretable. These tables need statistics and post hoc contrasts. In addition, their clinical names of the groups need to be on these tables.

8. Lines 368 through 392 about rurality contain numerous redundant sentences. This paragraph can be reduced by 50%.

9. The Discussion section would benefit from subheadings that would assist the reader to understand the key findings from this paper.

10. One of the goals of this type of study is to identify risk factors for a higher or worse symptom burden. Both the results and discussion sections of the paper should be written with this point of view.

Author Response

Thanks very much for the thoughtful, comprehensive suggestions. We very much appreciate this reviewer's comments. Our responses are below: 

The authors have been partially responsive in their revision of this manuscript. As stated previously, this paper is an important one - but several areas exist that warrant careful consideration to make this paper easy to read and clinically meaningful.

Response: Thanks very much for the careful and thoughtful review. We have made additional revisions to address the concerns raised.

1. The paragraph that was added to the introduction (lines 47 to 59) contains incorrect information. This paragraph really is not necessary. This paper is NOT evaluating symptom clusters [i.e., how symptoms group together]. It is evaluating how PATIENTS group together. The last two sentences of this paragraph are confusing and not stated correctly.

Response: Thank you for this comment. We agree and have removed this section.

2.  Lines 184 to 187 are not clear. In order for these sentences to have clarity, the authors need to define what they mean by symptom burden. The assumption of latent variable modeling - is that you are finding patients with distinct symptom profiles - what do the authors mean by symptom burden in this context? How did they determine symptom burden to be able to substantiate this sentence in the methods section of the paper?

Response: We agree and have made edits to clarify this point.

3. This reviewer suggested that the cohorts and the profiles have clinical names. However, none of this information was changed in the tables or figures. Use the clinical names throughout the paper and on all of the tables and figures.

Response: We have added clinical names to the tables and figures.

4. On the figure legends the groups are called mild, moderate, and high. However, on the figures and in the text, the groups are not high but severe. Please correct these errors.

Response: Thank you. We have corrected this error.

5. The legends state that the prevalence is plotted (i.e., whether the symptom occurred or not). You cannot plot what has not occurred.

Thank you. We agree. We only plotted symptoms that occurred (this is noted in the legend; the category ‘none’ [score of 0 the ESAS-r] was not plotted.

6. In the results section, the authors name the classes using the terms frequent and severe. However, they did not assess frequency. This study assessed symptom occurrence and severity. The other dimensions of the symptom experience that were not assessed were distress and frequency (i.e., how often the symptom occurs within a specified time interval (e.g., daily, weekly, monthly). This language needs to be corrected.

Response: Thank you. We have made this change and instead of frequent, we have used ‘prevalent’ throughout.

7. The shading on Tables 5 and 6 are not interpretable. These tables need statistics and post hoc contrasts. In addition, their clinical names of the groups need to be on these tables.

Response: Thank you for this suggestion. We note that we were not testing a priori or post-hoc hypotheses about the differences between the latent classes but rather aimed to provided descriptive information about the latent classes. We would be happy to include the MPlus output as supplementary information if this would be helpful.

8. Lines 368 through 392 about rurality contain numerous redundant sentences. This paragraph can be reduced by 50%.

Response: Thanks for this feedback. We added content regarding rurality as suggested by the other reviewer.

9. The Discussion section would benefit from subheadings that would assist the reader to understand the key findings from this paper.

Response: We have added headings to the Discussion.

10. One of the goals of this type of study is to identify risk factors for a higher or worse symptom burden. Both the results and discussion sections of the paper should be written with this point of view.

Response: We have made modifications to the Discussion section that we hope address this comment. Our analyses were exploratory and we have amended the aims to reflect this.

Reviewer 2 Report

Comments and Suggestions for Authors

The manuscript is greatly improved after incorporating the suggestions. I recommend accepting in present form.

Author Response

We are very grateful to the reviewer for their time and thoughtful comments. We very much appreciate their positive feedback. 

Round 3

Reviewer 1 Report

Comments and Suggestions for Authors

I want to congratulate the authors for their steadfast responses to the series of reviews of this paper. All of my comments were addressed in this version of the paper.